# Air Pollution in Moscow Megacity: Data Fusion of the Chemical Transport Model and Observational Network

**Nikolai Ponomarev** [1,2]*, **Vladislav Yushkov** [1] **and Nikolai Elansky** [2]

1   Physics Faculty, Lomonosov Moscow State University, Leninskie Gory, 1/2, Moscow 119991, Russia; yushkov@phys.msu.ru
2   Obukhov Institute of Atmospheric Physics, Russian Academy of Sciences, Pyzhevsky, 3, Moscow 119017, Russia; elansky@ifaran.ru
*   Correspondence: na.ponomarev@physics.msu.ru; Tel.: +7-916-250-9172

**Abstract:** Comparisons of observational data obtained at the Moscow Ecological Monitoring network (MEM) with numerical simulations using a chemical transformation and transport model (SILAM—System for Integrated modeLling of Atmospheric coMposition) showed that the errors in determining the gaseous pollutant concentrations in the urban atmosphere have a more complex structure than those assumed under the conventional algorithms of data assimilation. These errors are statistically nonstationary; they show a pronounced diurnal cycle and a significant lifetime. The statistical features of errors in numerical calculations also depend upon the type of pollutants, i.e., the chemical reactions in which they participate. Our analysis showed that the simulation errors are not small: the ratios of calculated and measured concentrations (even for daily averages at all measuring stations) may vary in a wide range. For the chemically active pollutants, the intradiurnal error variations may reach 100%. The diurnal cycle of such errors was found to vary according to seasons (in our case, summer and winter). The analysis of statistical properties of the errors, including their temporal and spatial variability, allows one to correct and adequately forecast the air pollution in the metropolis area at lead times up to three days in advance.

**Keywords:** megacity; air quality; chemical transport model evaluation; error model; data fusing

**PACS:** 92.60.Sz; 92.60.Aa

## 1. Introduction

The numerical models of chemical transformation and transport of atmospheric pollutants from different sources, including megacities, have been significantly improved over the past decades [1–4]. However, the use of these models for quantitative estimates and air pollution forecasting still remains a challenge, because our knowledge of the intensity and composition of emissions or of turbulent transport characteristics remains somewhat approximate, especially for the significantly inhomogeneous urban medium [5,6]. The lack of regular network observations of chemically active gaseous pollutants also presents a considerable challenge.

These concentration fields that determine the air quality are, to a great extent, random in their behavior. The infrequent intense periods of complex measurements, such as the BUBBLE experiment [7], do not provide assurance that the estimated characteristics of pollutant emissions or turbulent mixing will not vary with variations of meteorological conditions, transport load, and both the market and industrial dynamics in megacities. In contrast to many Russian megacities, the Moscow metropolis possesses an efficient system of air quality monitoring called the Moscow Ecological Monitoring (MEM) network. Here, the basic pollutants ($CO$, $NO_2$, $NO$, and $O_3$); both saturated and unsaturated hydrocarbons; and $PM_{10}$ and $PM_{2.5}$ aerosol fractions have been continuously measured

for many years [8–11]. The measurements are conducted at different locations: highways, residential areas, districts with a large industrial load, city parks, and suburbs.

The chemical transport models, unlike the local monitoring data that often contain data gaps, allow one to calculate the continuous fields of concentrations throughout the entire atmospheric boundary layer. However, these fields are always smoothed, because, in these models, the unknown sources and sinks are represented by averages. Measurements reflect not only some average or expected concentrations but also the effects of many random interrelated factors: rapidly varying and unknown sources, the mesoscale dynamics, atmospheric turbulence flows, and different types of measurement errors.

Such models are especially needed to predict rapid air quality deteriorations [12]. The necessary information obtained prior to the occurrence of extreme situations makes it possible to minimize the consequences adverse for human health and for environmental conditions. An anomalous increase in pollutant concentrations in the megacity's atmosphere may be associated not only with unfavorable meteorological conditions but also with significant variations in pollutant emissions (whose intensity and sources may rapidly vary) into the atmosphere. This is clearly supported by the fact that simulation results may differ from measurement data by 100% and more. The analysis of errors showed that the remaining still significant systematic errors, especially in calculating the fields of chemically active pollutants, suggest that the we are still far from achieving an improvement in chemical transport models.

Data assimilation in weather forecast and chemical transport models is a conventional approach that has established methods and principles (OI—optimal interpolation, 3D-Var, 4D-Var and, KF—Kalman filtering). The basic principle of observational data assimilation implies their use in correcting the initial and boundary conditions as well as in the model's parameters of the model itself. In other words, observations are included in a feedback circuit within the numerical model.

Uncertainty concept should be understood as deviations caused by the lack of accurate data on the urban emissions, the errors due to improper model parameterizations, and the measurement errors. Although each of these uncertainties has its own nature and dynamics, separating them from each other is difficult due to the lack of information. However, their main feature is that they do not remain unchanged and have different lifetimes: some of them are fast and local, while others vary slowly and cover the entire metropolis. Their spectral features can be estimated only by comparing model fields with measurement data.

The approach proposed here is based upon combining data obtained from simulations and from observations averaged over 20 min intervals, and it allows one to analyze in detail the properties of errors, separating them into quasi-systematic (that is, slowly varying) errors, such as daily and seasonal variability, and rapidly varying in time and space errors of different origin, such as instrumental, associated with weather conditions, traffic load, industrial activities, and parameterization of photochemical processes.

This approach is closest to the Kalman filtering of errors [13,14] and a probabilistic analysis of the dynamic equations with random sources. Such analysis presents an independent and complicated task that should be isolated from the problems of numerical implementation of model dynamics. At a later stage, results of this analysis should certainly be taken into account in upgrading chemical transport models.

The available studies of simulations of trace gas contents in the Moscow atmosphere include one or two pollutants usually considered for short time intervals (see, for example, [12,15,16]). In this work, we considered the primary and reliably measured gaseous pollutants, namely, $CO$, $NO_2$, $NO$, and $O_3$ for two longer periods in winter and summer: 1–31 January and 1–31 July 2014. This gives one a general idea about the basics of air pollution in Moscow and the errors in its simulations, as well as about the differences in gaseous components, which are dependent upon their chemical properties, distribution, source strength, and other factors. Such information on errors of chemical transport models is also important for making management decisions.

## 2. A Chemical Transport Model

The SILAM v.5.2. chemical transport model [17] used in this study makes it possible to describe atmospheric dynamic processes using both diffusion and advection equations in the Eulerian or Lagrangian coordinates. We used the Eulerian description of pollutant transport. The model takes into account both dry and wet deposition and chemical interactions. The algorithms for solving the equations of advective transport and diffusion are described in [18,19]. In the model, the chemical pollutant transformations are obtained through eight chemico-physical transformation modules, which describe basic acid chemistry and secondary aerosol formation, ozone formation in the troposphere and the stratosphere, radioactive decay, aerosol dynamics in the air, and pollen transformations [20]. A detailed description of the model and the calculation algorithms used is given in [17,21,22].

Validating complex and multicomponent chemical transport models and estimating the actual rather than nominal pollutant emissions present the initial difficult step in applying these models for the assessment of the air basin state in a large city and for the forecasting of severe ecological situations. Another problem of dynamic simulation is the low spatial resolution of the models, which stays between one and ten kilometers. A cell of this size may simultaneously include several dissimilar landscapes, such as highways, parks, and dwelling zones. These objects may have distinct macroturbulence characteristics and their own intensity and regime of pollutant emission. Moreover, the situation is complicated by the presence of a large number of chemical bonds between gaseous components, which are only approximately parameterized in current models [20,23].

*Characteristics of Numerical Experiments*

The spatial resolution of SILAM's computational grid in our case was 0.10 degrees in longitude (about 8 km) and 0.05 degrees in latitude (about 8 km) at 69 altitudes for meteorology and 9 altitudes for chemical transformations. The grid area is a rectangle within 50.75–60.70° N and 32.60–42.50° E. Data of the on-line forecast calculated with the HIRLAM model [24] at a time resolution of 3 h were specified as initial meteorological fields.

The initial and boundary conditions for pollutant concentration fields were specified based on pollutant fields calculated at the Finnish Meteorological Institute using the SILAM model for Europe and the European Russia [25]. Most air pollutant emissions were taken from the TNO-2011 Inventory data [26]. The CO and NOx emissions within the territory of the Moscow megacity were specified based upon annual emission values provided in [10]. Analysis of data obtained at the MEM network and numerical experiments on the optimization of urban air pollution sources described in [27,28] allowed us to distribute them over time with one-hour time step and across Moscow territory. The CO and NOx emissions outside Moscow were taken from the TNO-2011 Inventory data. The emissions of non-methane volatile organic compounds (NMVOCs) were obtained from the MACCity inventory [29].

The months of January and July in 2014 chosen for calculations are of interest for several reasons. Firstly, this allowed us to detect differences between winter and summer variations of pollutant concentrations within the Moscow's atmosphere. Secondly, both months include two-week periods with the presence of a stable airmass (winter and summer blockings). The wind velocity in a stable airmass is low, which is one of the reasons for accumulation of atmospheric pollutants that are hazardous to human health. Finally, a sufficiently long period covered with computations allowed us to obtain statistically reliable estimates.

## 3. Observational Data and Processing Methods

The observation data were taken from the database of the MEM network belonging to the State Environmental Protection Institution "MosEcoMonitoring". The MEM stations rather uniformly and densely cover the entire territory of Moscow [30]. The network also includes a few out-of-city stations that produce information on the contents of pollutants in both windward (regional background) and leeward regions. In this work, we used the mea-

surement data on the *CO*, *NO*, $NO_2$, and $O_3$ concentrations averaged over 20 min intervals (see Table 1). These and other pollutants were measured with instruments recommended by the World Meteorological Organization (WMO) for use by the Global Atmospheric Watch network (WMO GAW). The instruments were regularly calibrated according to international standards and calibration gas mixtures. In our previous work [10], observational data obtained for 2005–2014 were analyzed, and the annually mean *CO* and $NO_x$ emissions were evaluated.

The task of systematization and analysis of the quality of observational data obtained at the network stations that monitor the atmospheric pollution is not easy. The multiyear data series contain a significant number of gaps caused by the necessary periods of calibrations of instruments and by their occasional failures. Using new instruments with different technical characteristics affects the measurement accuracy and the homogeneity of data series. The local spectral properties of errors in such measurements vary with station locations and with the amount of data gaps. Such numerous uncertainties raise the questions of observation quality: what is the relationship between the errors at different measurement points? What are the spectral properties of measurement errors, and is it possible to separate them from simulation errors?

*Processing the Initial Series of Measured Pollutant Concentrations*

The initial series of measured *CO*, *NO*, $NO_2$, and $O_3$ concentrations were selected individually for each station for the time intervals corresponding to our computations and compared with simulation results interpolated to respective measurement points. This made it possible to estimate the quality of instruments and to detect some differences in the sensitivity of instruments used for measuring the *NO* and $O_3$ concentrations; comparing measured and calculated values will show the differences between the minima of their diurnal variations.

The criterion for selecting the stations was the uninterrupted operation of sensors during 90% of time for the periods under consideration. It should be noted that the number of stations that measure ozone on a regular basis is small. Therefore, the statistical averaging over a small number of stations results in significant sampling errors as compared to that of other pollutants and leads to errors in estimating the error distribution parameters: the median value and the interquartile range.

Table 1 gives the *CO*, $NO_2$, *NO*, and $O_3$ concentrations averaged over January and July 2014 according to observations at stations with less than 10% of missing observations and model (SILAM) calculations in relation to the location of the selected stations. The mean *NO* (and *CO* for July) concentrations calculated using the SILAM model are noticeably lower than respective measured values, which may occur due to the fact that high concentrations of some pollutants cannot be reproduced with a mesoscale model for measurement points situated close to highways, which are characterized by high local concentration values.

To decrease the random spatial component of errors in comparing measured and simulation data, the average concentrations over all observation points, for which a sufficient amount of data (for selected time series) was available, were calculated for each gas component as

$$\langle q(t_j) \rangle = \frac{1}{N} \sum_{i=1}^{N} q_{ij}. \tag{1}$$

here, $q_{ij}$ is the pollutant concentration measured at the *i*-th station at the *j*-th time and *N* is the total number of stations whose data were used in averaging over the city area. The calculations were performed for the measurement data denoted below by $\langle q_M \rangle$ (MEM) and for model fields $\langle q_S \rangle$ (SILAM), whose values were interpolated to measurement points.

The Kolmogorov–Zurbenko filter [31], capable of working with data series, including those that contain gaps, was used to isolate the intradiurnal and interdiurnal components

in the time series of concentrations. This filter may be defined as k times iteration of a moving average (MA) filter with an averaging window with a width of $2m + 1$ points:

$$y_j = \frac{1}{2m+1} \sum_{k=-m}^{m} q_{j+k},$$ (2)

where $q_{j+k}$ is the initial pollutant concentration in time series and $y_j$ is the averaged concentration. The reapplication of the MA filter to already smoothed data makes it possible to obtain a steeper spectral transmission window characteristic than that of a simple moving-average filter ( $\text{sinc}^k(\frac{2\pi}{m}i)$ ). At each subsequent iteration, the $y_j$ values obtained at the previous step are taken as a series of $q_j$ values. We used three iterations in this work.

**Table 1.** Number of all the Moscow Ecological Monitoring network (MEM) stations and only those participating in comparisons with the SILAMcalculation results (less than 10% of missing observations) and measured and calculated pollutant concentrations (mg/m$^3$) averaged over periods of 1–31 January and 1–31 July 2014 for the selected stations.

| Pollutant | January | | | | July | | | |
|:---:|:---:|:---:|:---:|:---:|:---:|:---:|:---:|:---:|
| | Number of Sites | >90% of Data | MEM Mean Conc. | SILAM Mean Conc. | Number of Sites | >90% of Data | MEM Mean Conc. | SILAM Mean Conc. |
| CO | 37 | 24 | 0.46 | 0.47 | 39 | 19 | 0.53 | 0.34 |
| $NO_2$ | 34 | 25 | 0.033 | 0.030 | 35 | 28 | 0.030 | 0.031 |
| NO | 34 | 21 | 0.033 | 0.022 | 34 | 21 | 0.022 | 0.012 |
| $O_3$ | 9 | 4 | 0.016 | 0.014 | 11 | 7 | 0.045 | 0.043 |

The presence of a clearly pronounced diurnal cycle of pollutant concentrations averaged over the megacity generates a need for estimating errors in numerical calculations of not only high daytime concentrations of pollutants but also their low night-time concentrations. Thus, a quite admissible error of 10–20% under daytime conditions may lead to a relative error of over 100% in the night-time if the value of simulation error is considered constant. On the other hand, if the model error were only systematic, although varying with time, it would be easily taken into account as if this error were quite random and did not correlate with its previous value. However, in practice, both assumptions are very rough, and, for their improvement, a detailed spectral analysis of the characteristics of random variations in measured pollutant concentrations and simulation errors is necessary.

## 4. Comparing Observational Data with Results of Numerical Experiments

### 4.1. Space and Time Variations in Model Calculation Errors

The main feature that distinguishes between spatial pollutant distributions obtained from observational data and calculations using the chemical transport model is the presence of significant local inhomogeneities in observational data, which cause a high level of local errors or, more correctly, natural variations. Comparing between the time series of observations and model estimates interpolated to observation points demonstrates an example of such local differences (Figure 1a).

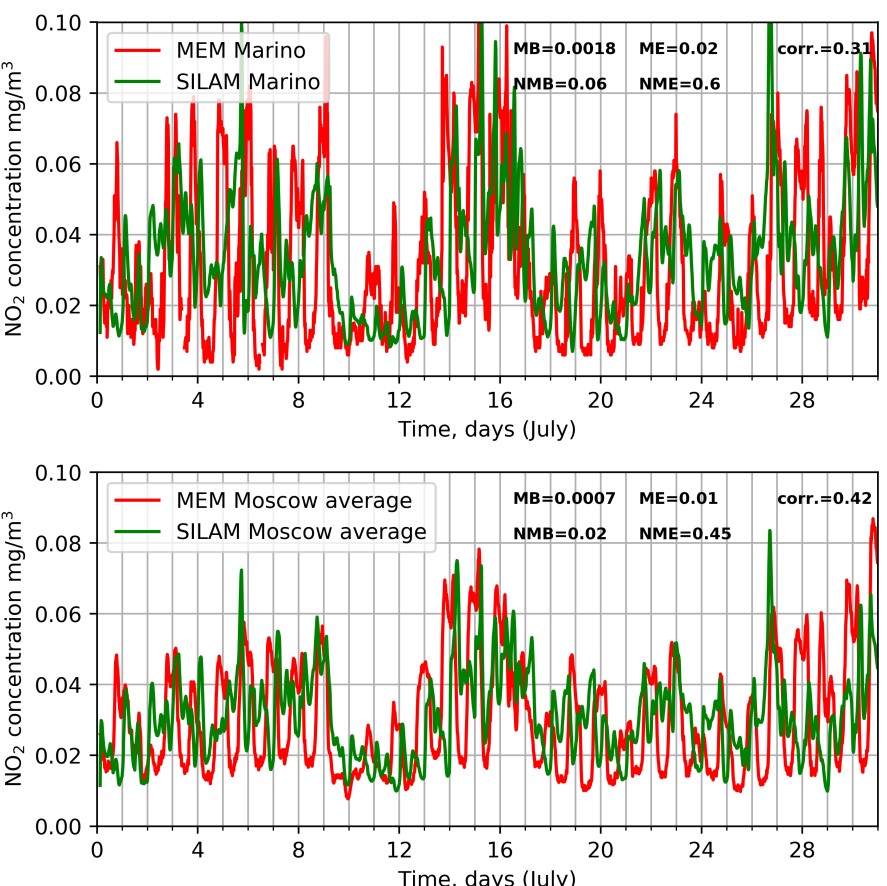

**Figure 1.** Local errors and the errors of average concentrations (MB—mean bias, NMB—normalized mean bias, ME—mean error, NME—normalized mean error, corr—correlation coefficient). **Top**: the $NO_2$ concentrations measured on 1–31 July 2014, and calculated using the SILAM model for the same period for the Mar'ino station. **Down**: the $NO_2$ concentrations averaged over the selected MEM stations for a period of 1–31 July 2014, according to measurement data and numerical calculations.

After the pollutant concentrations averaged over many stations, this conditional average over the megacity shows a significantly better agreement with simulation results (Figure 1b). Variations in the amplitude of the diurnal cycle due to varying meteorological conditions (mean wind velocity, intensity of convective mixing in the atmospheric boundary layer, passage of cyclones or anticyclones over the megacity, and change in air masses with different chemical histories) are traced significantly better.

Note that the most chemically active pollutants $NO$ and $O_3$ show less agreement with observational data. As is shown below in Section 4.4, the calculated and measured time series of the $CO$ and $NO_2$ concentrations averaged over the megacity show the best agreement for both summer and winter seasons.

Although deviations of pollutant concentrations in both calculated and measured data still may reach tens of percent, the regularity in the behavior of the diurnal cycle of average (over Moscow) concentration allows one to correct simulation errors according to the data on the average diurnal error cycle (see below in Section 4.4).

### 4.2. Local Variation Distribution Functions

The spatial inhomogeneity and temporal variability of differences between calculated (using the model) and measured pollutant concentrations may be considered as random because there are no data on what causes these deviations beyond the scope of the model. If we assume that the average (over the megacity) pollutant concentrations describe the

main regularities of time variations in chemical pollutants, then the ratio between the local and average (over Moscow) concentrations may be considered as a random variable.

The median of the probability distribution of such a characteristic is bound to be close to 1, while the interquartile range may be high. To demonstrate this, the empirical probability distribution functions of this normalized characteristic (Figure 2) were constructed on a semi-logarithmic scale. They were constructed for the set $\{\frac{q_{ij}}{\langle q \rangle_j}\}$, where $i = \overline{1, N}$, $j = \overline{1, M}$, $N$ is the total number of stations at which the measurements were taken, and $M$ is the number of 20-min time intervals within the calculation period. The logarithmic scale of the abscissa axis allows one to see that deviations from the average (over Moscow) concentration values may amount to hundreds of percent in some cases. One can also see that deviations from the average in measurement data significantly exceed the local variations according to model calculations.

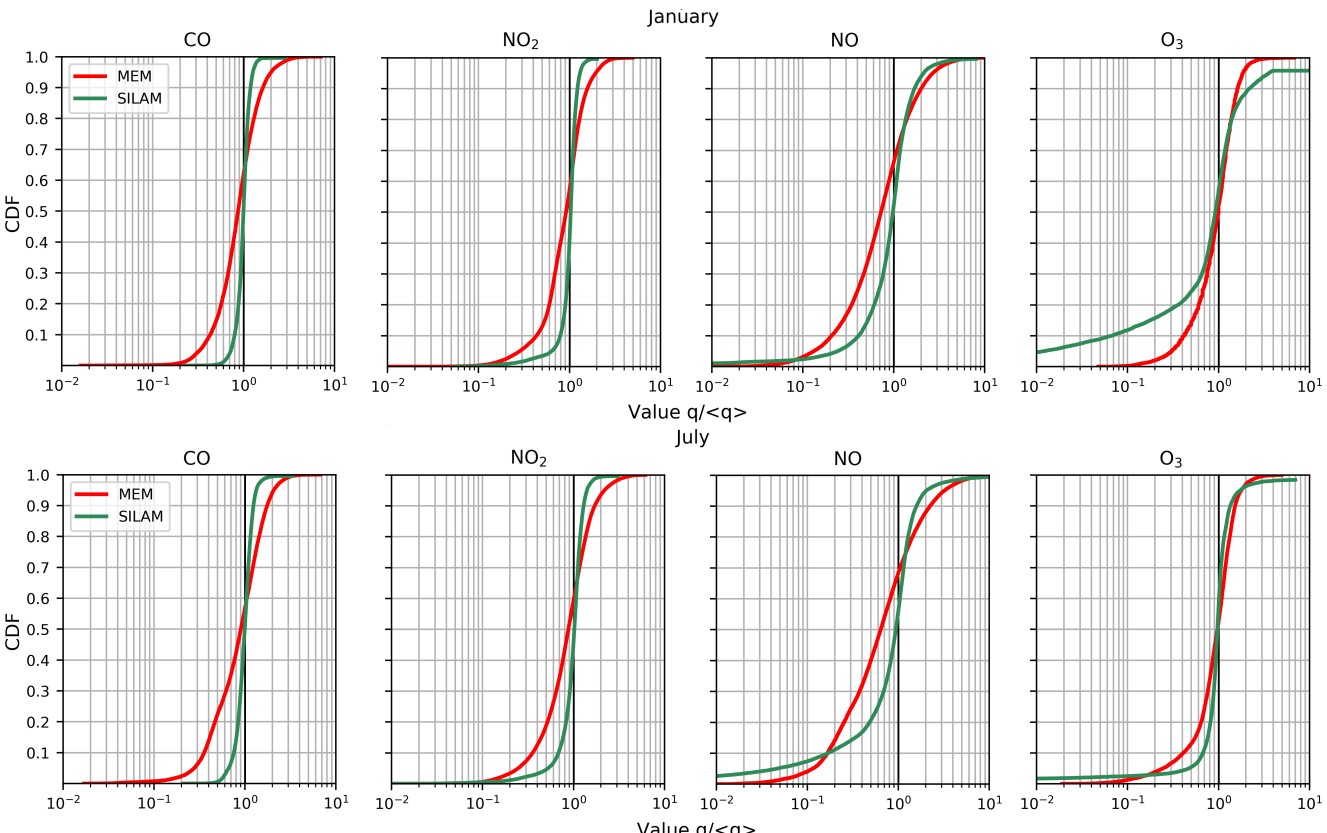

**Figure 2.** Distribution functions for the normalized values $q/\langle q \rangle$ of the *CO*, *NO*, *NO$_2$*, and *O$_3$* concentrations calculated using the SILAM model (green line) and measured at the MEM stations (red line) for both the winter (1–31 January 2014, top row) and summer (1–31 July 2014, bottom row) periods.

The parameter characterizing the range of local variations is the relative interquartile range $Q$—the ratio between the third quartile $P_{75}$ (the value below which there are 75% of concentration values in the sample data) and the first quartile $P_{25}$ (the value below which there are 25% of concentration values in the sample data):

$$Q = P_{75}/P_{25} \,. \tag{3}$$

The quartile values were determined individually for each sampling (that is, empirical) cumulative distribution function (CDF) of normalized characteristic. Table 2 displays medians and interquartile ranges for January and July for each of the four pollutants according to the measurement and simulation data. The difference $P_{25} - P_{75}$ shows within what range (with regard to the scatter) half of the entire observational data sample is

located. The long-lived *CO* and $NO_2$ pollutants have a smaller interquartile range, which implies that they are distributed over space more uniformly and closer to the average (over the city) concentration.

**Table 2.** Median, quartiles (25 and 75%), and relative interquartile range Q for the ratio between current 20-min local concentrations at the MEM stations and their values averaged over all stations and similar calculations using the SILAM model for the pollutant concentrations interpolated to the points of station locations for periods of 1–31 January and 1–31 July 2014.

| Pollutant | Period | Median | | 1st Quartile | | 3rd Quartile | | Q | |
|---|---|---|---|---|---|---|---|---|---|
| | | MEM | SILAM | MEM | SILAM | MEM | SILAM | MEM | SILAM |
| CO | January | 0.87 | 1.00 | 0.63 | 0.91 | 1.20 | 1.09 | 1.91 | 1.20 |
| | July | 0.90 | 0.99 | 0.54 | 0.87 | 1.32 | 1.12 | 2.44 | 1.28 |
| $NO_2$ | January | 0.92 | 1.03 | 0.65 | 0.93 | 1.20 | 1.12 | 1.86 | 1.20 |
| | July | 0.87 | 1.01 | 0.58 | 0.87 | 1.25 | 1.14 | 2.15 | 1.31 |
| NO | January | 0.72 | 0.98 | 0.41 | 0.69 | 1.22 | 1.25 | 3.00 | 1.80 |
| | July | 0.64 | 0.93 | 0.31 | 0.60 | 1.21 | 1.20 | 3.92 | 2.02 |
| $O_3$ | January | 1.00 | 0.89 | 0.68 | 0.49 | 1.30 | 1.21 | 1.91 | 2.47 |
| | July | 0.97 | 0.97 | 0.71 | 0.86 | 1.25 | 1.10 | 1.77 | 1.28 |

Another important feature of the measurement data distributions, which is mostly characteristic of *NO*, is the deviation of the median of relative distribution from 1. In other words, the distribution of errors in calculating the *NO* concentrations is not Gaussian even approximately and the contribution of a smaller number of high concentrations (exceeding the average), for example, in the vicinities of highways, prevails over that of a larger number of low concentrations (below the average).

Some stations (less than 10% of the total number) show significant tenfold (and even higher) deviations of both instantaneous and local values from the average over the city. One can assume that these deviations from the average are not just "accidental" but may be associated with some features of the operation or location of these stations. In other words, such empirical distribution functions allow one to select stations or to perform their clusterization by location or infrastructure types (highways, industrial zones, residential areas, forest parks, and others). Exciting a chemical transport model with random (but statistically close to measurements) sources/sinks, as in meteorology, in the form of nudging may also be regarded as one more application of a statistical approach [32,33].

On the whole, for these four pollutants, the higher *Q* values in the measurement data support the hypothesis of the spatial smoothness of the concentration fields calculated with models. Since these model fields cannot reproduce locally measured features, they should be compared using averaged estimates such as medians.

*4.3. Spatial Pattern of Model Errors*

The spatial character of local uncertainties in model concentration estimates can be expressed in the form of a ratio between simulated and measured concentrations $q_S/q_M$. Since this quantity presents a random function of time, regularities in the spatial distribution of deviations of model estimates from measured ones are expressed through statistical distribution characteristics of this quantity, for example, median and interquartile range. Such maps for both winter and summer seasons are shown in Figure 3. The median of the distribution of ratios between calculated and measured concentrations over the calculation periods under consideration—1–31 January and 1–31 July 2014—is denoted by marker colors. If the median of distribution is higher than 1, the model mostly overestimates pollutant concentrations at a given point, and if the median is lower than 1, the model mostly underestimates real pollutant concentrations.

The relative interquartile range $Q_r$, i.e., the ratio of the third quartile of distribution to the first one, was calculated to characterize time domain variations of local errors or uncertainties in model estimates for each observation point. All stations were divided into three groups according to $Q_r$ values. Group 1 (triangle) corresponds to the first quantile (25%) of the data sample including $Q_r$ values for each observation point, group 2 (circle) corresponds to both the second and third quantiles (25–75%), and group 3 (square) corresponds to the fourth quantile (75–100%). In other words, the measurement data obtained at the stations of group 1 have the lowest random variability $q_S/q_M$, and the measurement data obtained at the stations of group 3 have the highest random variability. This implies that, for the stations of group 3, the forecast even if adjusted for an average relative error will rarely agree with measurement data, because it contains a significant random component.

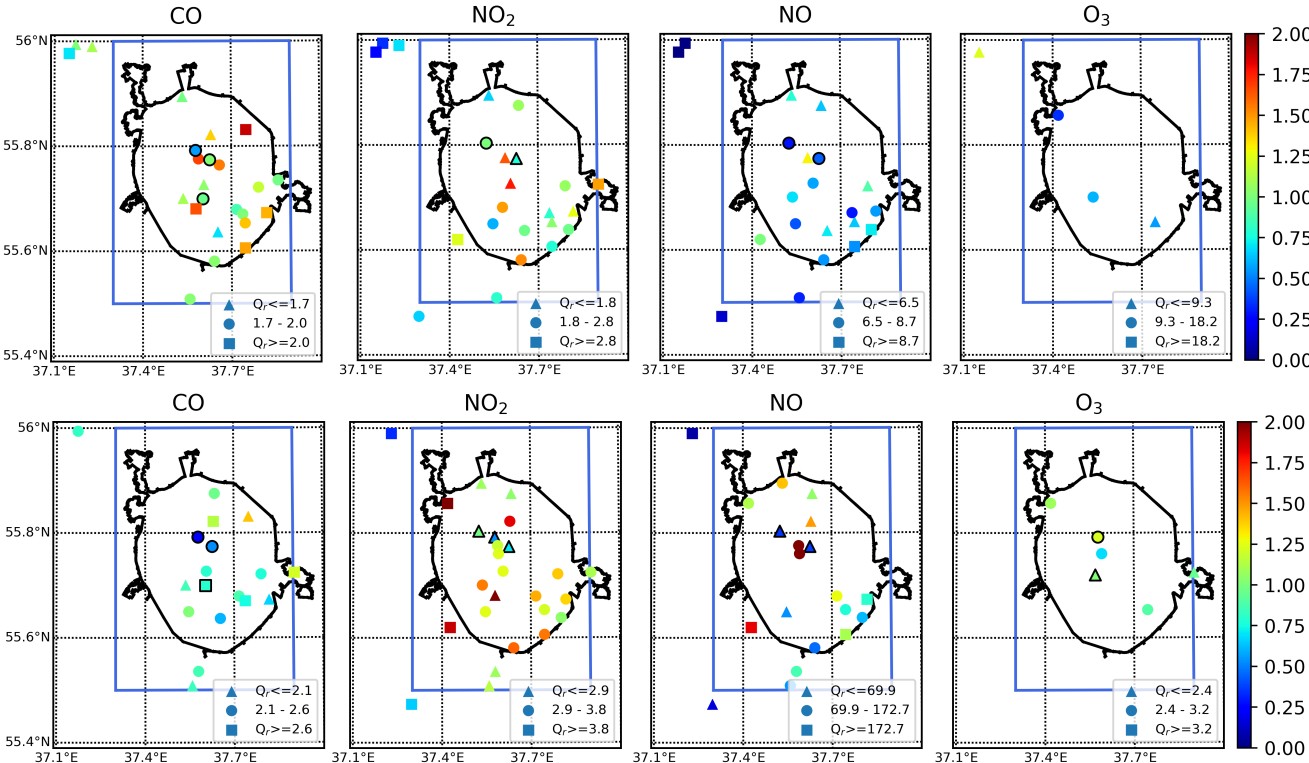

**Figure 3.** Spatial distribution of statistics of calculated and measured (at the MEM network) concentrations of $CO$, $NO$, $NO_2$, and $O_3$ for 1–31 January (top row) and 1–31 July 2014 (bottom row). The characteristic value of the relative interquartile range $Q_r$ at each station is denoted by marker types (triangle, circle, and square). The area for which the $CO$ and $NO_x$ emissions were specified based upon analysis of data obtained at the MEM network and from numerical experiments on the optimization of urban air pollution sources is marked by blue rectangle.

As might be expected, the long-lived $CO$ and $NO_2$ pollutants yield the best results, i.e., the values that are close to 1. These pollutants also have the smallest relative interquartile range. The stations located near highways are marked by black outlines. For these stations, the ratio $q_S/q_M$ for $CO$, $NO$, and $NO_2$ proved to be less than one with a small interquartile range $Q_r$. In addition, for $O_3$ at the stations in vicinities of highways, $q_S/q_M$ is larger than one, which may be explained by the effect of motor transport, because a large amount of $NO$ emitted by motor transport reacts with $O_3$ and, thus, significantly decreases its concentration in vicinities of highways.

An important feature of spatial variations in disagreements between modeled and measured concentrations is their rapid and random alternation over the city map, which reflects the mosaic alternation of highways and industrial enterprises with recreation zones and residential areas. Table 3 gives the distribution of the groups of stations according to

the $q_S/q_M$ quartiles and the types of urban infrastructure. It can be observed in Table 3 that both residential areas and highways are almost equally distributed by the level of model errors. Thus, one may conclude that, within the framework of a chemical transport model with a spatial resolution of 1–10 km, it is hardly possible to accurately describe measured concentrations at observation points. Therefore, we can obtain a regular pattern of Moscow's air pollution and errors in simulating this pollution only by averaging these concentrations over a large number of observation points.

**Table 3.** Distribution of relative $q_S/q_M$ errors at the stations located in vicinities of different infrastructure objects according to relative quartile $Q_r$ values.

| Pollutant | Set | Residential | | Highways | | Mixed | | Natural | |
|---|---|---|---|---|---|---|---|---|---|
| | $Q_r$ | January | July | January | July | January | July | January | July |
| CO | 1 (<25%) | 4 | 1 | 0 | 0 | 2 | 3 | 1 | 1 |
| | 2 (25–75%) | 2 | 1 | 3 | 2 | 6 | 6 | 0 | 1 |
| | 3 (>75%) | 2 | 1 | 0 | 1 | 2 | 0 | 1 | 0 |
| $NO_2$ | 1 (<25%) | 3 | 1 | 1 | 3 | 3 | 4 | 0 | 0 |
| | 2 (25–75%) | 0 | 3 | 1 | 0 | 11 | 8 | 0 | 1 |
| | 3 (>75%) | 3 | 2 | 0 | 0 | 2 | 4 | 0 | 0 |
| NO | 1 (<25%) | 1 | 0 | 0 | 2 | 5 | 4 | 0 | 0 |
| | 2 (25–75%) | 1 | 4 | 2 | 0 | 6 | 6 | 0 | 0 |
| | 3 (>75%) | 2 | 0 | 0 | 0 | 3 | 5 | 0 | 0 |
| $O_3$ | 1 (<25%) | 1 | 0 | 0 | 1 | 0 | 0 | 1 | 0 |
| | 2 (25–75%) | 1 | 2 | 0 | 1 | 1 | 1 | 0 | 0 |
| | 3 (>75%) | 0 | 0 | 0 | 0 | 0 | 1 | 0 | 0 |

*4.4. Seasonal Distribution of Model Errors*

At some stations, comparison between the measured and calculated (using the SILAM model for the same points) concentrations shows a significant random disagreement between them. The averages of many stations' pollutant concentrations are in a significantly better agreement (in terms of conventional RMS norm) with respective average model values. However, these averaged concentrations may also demonstrate noticeable disagreements, especially for concentrations of chemically active and short-lived pollutants. Therefore, the analysis of different regularities in simulation errors (even for concentrations averaged over the megacity) requires the knowledge of respective probability distribution.

In order to compare the average (over all stations) concentrations obtained from observations $\langle q_M \rangle$ (MEM) with calculations $\langle q_S \rangle$ (SILAM), Figure 4 shows the CDFs for their ratios for the summer and winter periods on a semilogarithmic scale. The average error of SILAM calculations is closer to that for $CO$ in winter and for $NO_2$ in summer (see below in Section 4.6). Even for the long-lived pollutants, approximately in 10% of cases, the average observed concentrations exceed or underestimate calculated values by a factor of 1.5 to 2.

The medians of error distribution in concentrations averaged over the city are noticeably close to one, which implies the adequacy of the SILAM model for air pollution forecasts for the whole city. For the concentrations of the short-lived and chemically active $NO$ and $O_3$ pollutants, the difference between their modeled and measured concentrations averaged over Moscow is more significant in both winter and summer periods. In both summer and winter, the model significantly (by an order of magnitude) underestimates the average (over Moscow) current (20-min) concentration of $NO$. This happens in about 30% of cases. The distribution of these errors in the summer period significantly differs from their winter time distribution. The width of the distribution function for the relative errors

in *NO* and $O_3$ concentrations (interquartile range) also proves to be an order of magnitude larger when compared to the *CO* and $NO_2$ distributions.

The high values of the current 20-minute (not averaged over a long time interval) error in Moscow average concentrations of chemically active pollutants imply that at least some part of it may be systematic and nonrandom and may vary during the day and between the seasons.

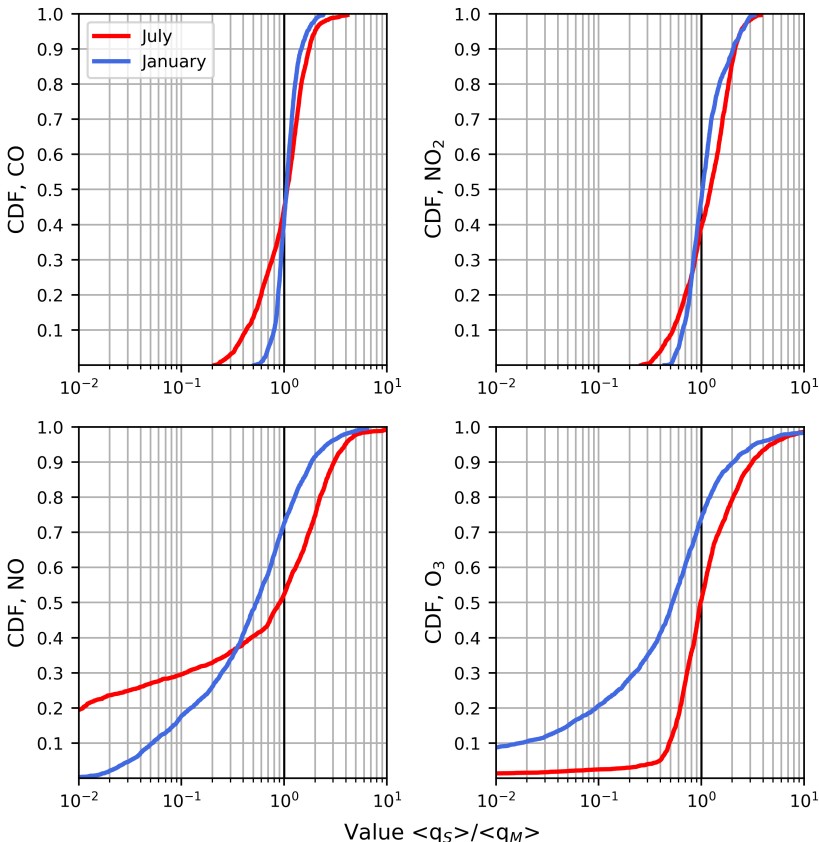

**Figure 4.** Distribution function for the ratios between average (over Moscow) concentrations (with averaging over 20 min intervals) calculated using the SILAM model and measured at the MEM stations: $\langle q_S \rangle / \langle q_M \rangle$ for *CO, NO, $NO_2$*, and $O_3$ for 1–31 January (blue line) and 1–31 July (red line), 2014.

### 4.5. Intradiurnal Cycle of Model Errors and Its Correction

Since photochemical processes, emissions, dry deposition, and turbulent mixing have a clearly pronounced diurnal cycle, it is precisely these diurnal simulation error variations that show, to the full extent, the weak points of any chemical transport model. Although the analysis of such variations does not show the cause of errors, recommendations that follow from such statistical analysis for improving the models should lead to changes in the ratio $\langle q_S \rangle / \langle q_M \rangle$ between the SILAM and MEM averages over Moscow and bring this ratio closer to one. Figure 5 shows the monthly averaged diurnal cycle of this ratio for each of the four gaseous pollutants for winter and summer.

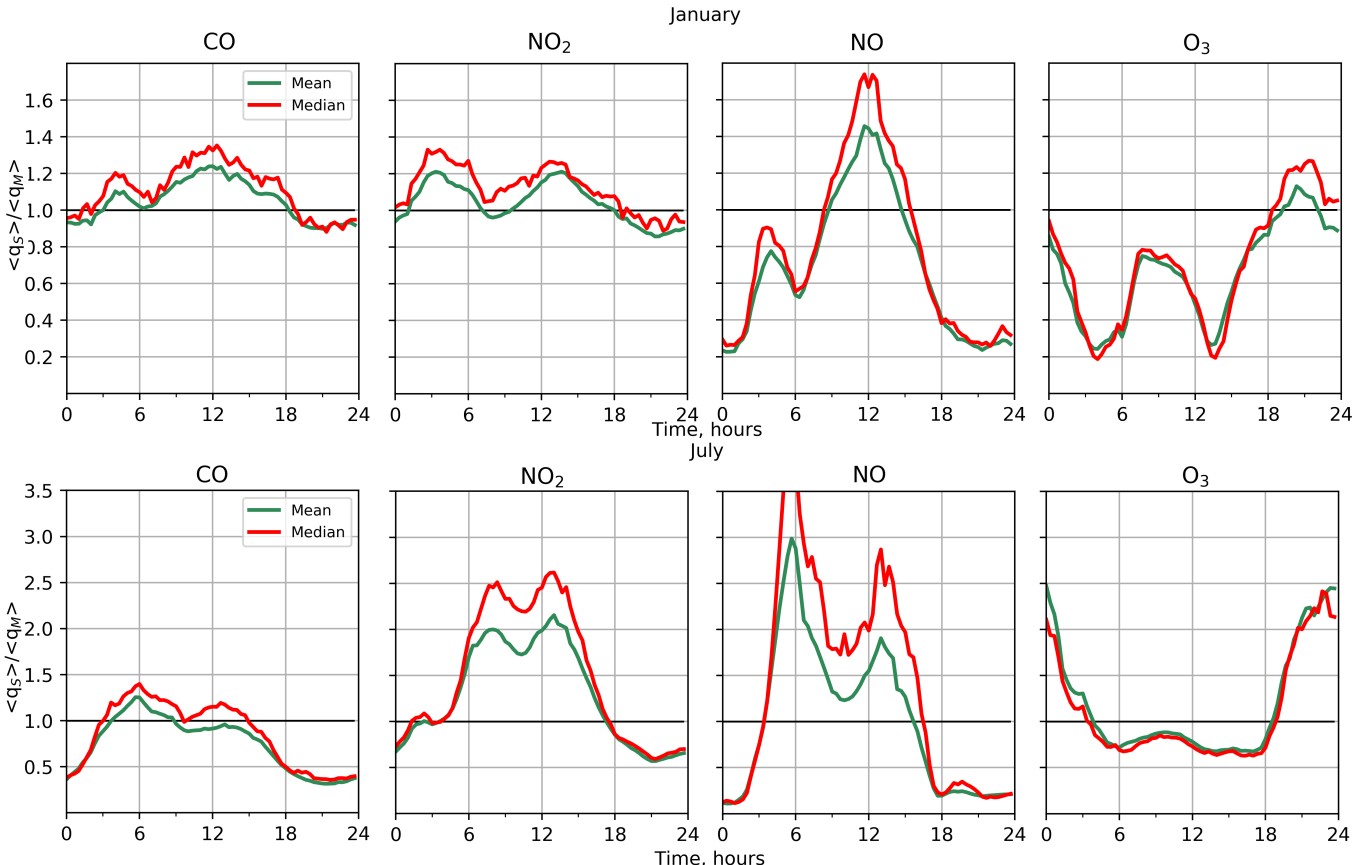

**Figure 5.** The diurnal cycle of the ratio between the average (over Moscow) *CO*, *NO*, *NO₂*, and *O₃* concentrations calculated with the SILAM model and measured at the MEM stations: $\dfrac{\langle q_S \rangle}{\langle q_M \rangle}$ (red line) and, similarly, the median of this ratio (green line) in winter (1–31 January 2014, top row) and in summer (1–31 July 2014, the bottom row). Time is in UTC.

First, we note that the diurnal cycle of the ratio between average concentrations over the megacity is significant according to both model calculations and to field observations. This ratio for chemically active pollutants may reach 3 (an error of 200%). At the same time, we note that the diurnal cycle of this ratio widely varies depending on season according to model calculations and observations. This result shows how the model quality is generally difficult to estimate. Even if it adequately reproduces the diurnal cycles of the *CO* and *NO₂* concentrations, it does not necessarily mean that this model is capable of reproducing diurnal cycles of other important components of atmospheric pollution. Moreover, verification of the model on the basis of measurement data obtained during only one month does not allow one to estimate its error in other seasons.

During the winter period, when the content of *NO* is usually low, its calculated concentration may be 2–5 times lower than the measured concentration; an exception occurs over the time interval from 10:00 to 14:00 UTC (14:00–18:00 LT in 2014), when the ratio between calculated and measured concentrations is close to or exceeds one. From 17:00 to 04:00 UTC (21:00–08:00 LT), when the concentration of nitrogen oxide is significantly underestimated by the model, the content of ozone is overestimated and reaches 1.1 at 20:00 (00:00 LT). During the summer period, the situation changes quite significantly: the *NO* concentration is noticeably (two–three times) overestimated when compared to observations from 04:00 to 16:00 (08:00–20:00 LT). During the same period, ozone is underestimated by about 1.5 times.

If we assume that the chemical transport models better forecast only average diurnal (but not 20-min) pollutant concentrations, then the obtained curves may serve as correction functions for numerical prediction. Multiplying the model intradiurnal cycle by these cor-

rective functions, we will obtain concentrations that are, on average, closer to observations and certainly without considering the remaining random variability component.

### 4.6. Long-Term Quasi-Systematic Errors, Their Forecast, and Kalman Filtering

The lower frequency that persists for a day or longer and presents almost systematic errors is associated with variations in industrial emissions, neglected sources (such as accidents on solid domestic waste (SDW) sites), and systematic errors in describing turbulent transport in the atmospheric boundary layer (ABL). These slowly varying simulation errors predicted for one or two days of the inertial forecast may be separated from intradiurnal variations using a simple low-frequency filter. Kolmogorov–Zurbenko filtering with the averaging scales of 12, 24, and 48 h was applied to the initial time series of measurement data obtained with a sampling interval of 20 min and through model calculations. In averaging with a window of 12 h or less, the intradiurnal variability begins to dominate the interdiurnal one, and, on a 48 h scale, error variations that can be predicted using measurement data for the previous day are strongly suppressed. Moreover, the average diurnal interval of our analysis naturally coincides with the interval between successive forecasts by the SILAM mesoscale model.

It can be observed in Figure 6 that the relative difference between the average measured and numerically calculated diurnal concentrations slowly varies with time and may be taken into account using either an autoregressive model (inertial forecast), or the Kalman filter [13]. It is clearly seen that the minimum error in daily average forecasts occurs for the longer lived $CO$ and $NO_2$ pollutants. It should also be noted that the model overestimated the concentration of $NO_2$ for the period of 1–13 January and underestimated it for the period of 14–31 January under the conditions of a stationary air mass over Moscow.

For estimating the Kalman filter parameters at each time step for their forecasting at a unit lead time, we used the following Gaussian linear model given in [34]:

$$\begin{cases} y_t = C_t x_t + \mu_t \\ x_{t+1} = A_t x_t + \nu_{t+1} \end{cases} \tag{4}$$

where $y_t$ is the concentration measured at time $t$, and $x_t$ and $x_{t+1}$ are the model estimates using the Kalman filter for times $t$ and $t+1$, respectively, while $C_t$ is the correlation coefficient between observed and measured concentrations, i.e., for the two data series each consisting of 31 values, and $A_t$ is the autocorrelation coefficient for calculations with the SILAM model and a one-day delay.

Then, the Kalman extrapolation and correction procedures were applied to the values determined in (4). The first procedure allows one to forecast an a priori estimate of the pollutant content $x(t)$ built on the basis of the knowledge of variations history in modeled concentrations, while the other procedure allows one to forecast the value of $x(t)$ a posteriori in considering the value of $y(t)$ newly measured at step $t$ and the updated estimates of their statistical properties. The mathematical formulations of these procedures and some examples of their application in geophysics may be found, for example, in [35,36].

Figure 7 presents the results of these calculations. Using such an approach to the measurement data series obtained in January and July 2014 led to evidently improved results forecasted by the model, especially for $CO$ and $NO_2$.

The summer periods of atmospheric blocking over Moscow are known to result in increased $CO$, $NO$, and $NO_2$ concentrations (see, for example, [37]). The influence of winter atmospheric blockings upon the urban atmosphere composition is understood to a lesser degree. Therefore, it is important to note that a blocking anticyclone that stayed for two weeks in January 2014 caused an increase in the average diurnal pollutant concentrations that reached their maximum on 20 January 2014. Its distinctive feature was the presence of a clearly pronounced concentration peak, which resulted in a derivative jump and a deterioration in the operation of the Kalman filter for $NO_2$. Therefore, for January 2014,

we tested the Kalman filter algorithm including a correction stage, but without the stage of extrapolation.

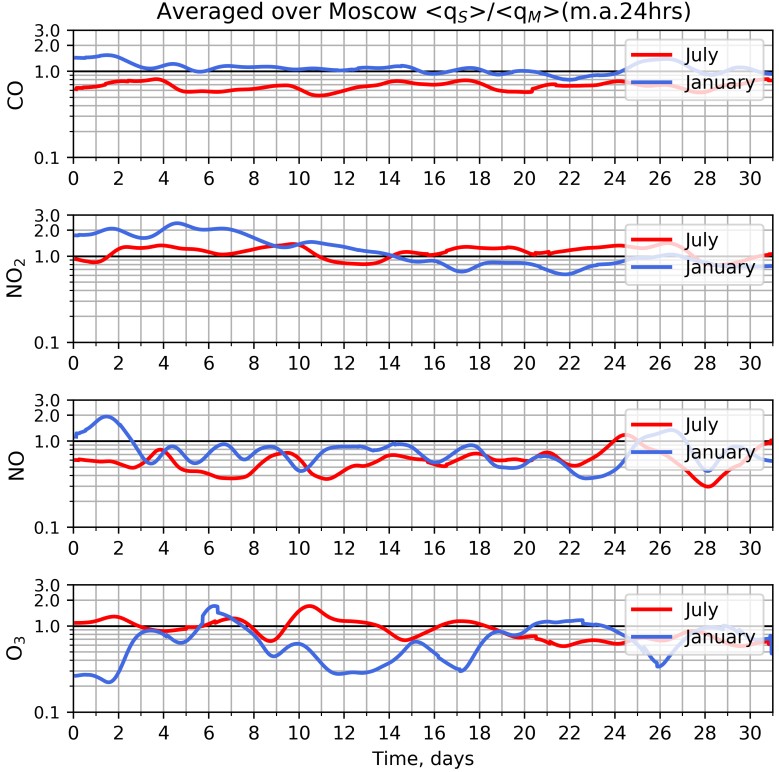

**Figure 6.** Ratio between the average (over Moscow) *CO*, *NO*, *NO₂*, and *O₃* concentrations calculated using the SILAM model and measured at the MEM stations for 1–31 January and 1–31 July 2014. The concentration data series were smoothed with the Kolmogorov–Zurbenko filter with an averaging window of 24 h.

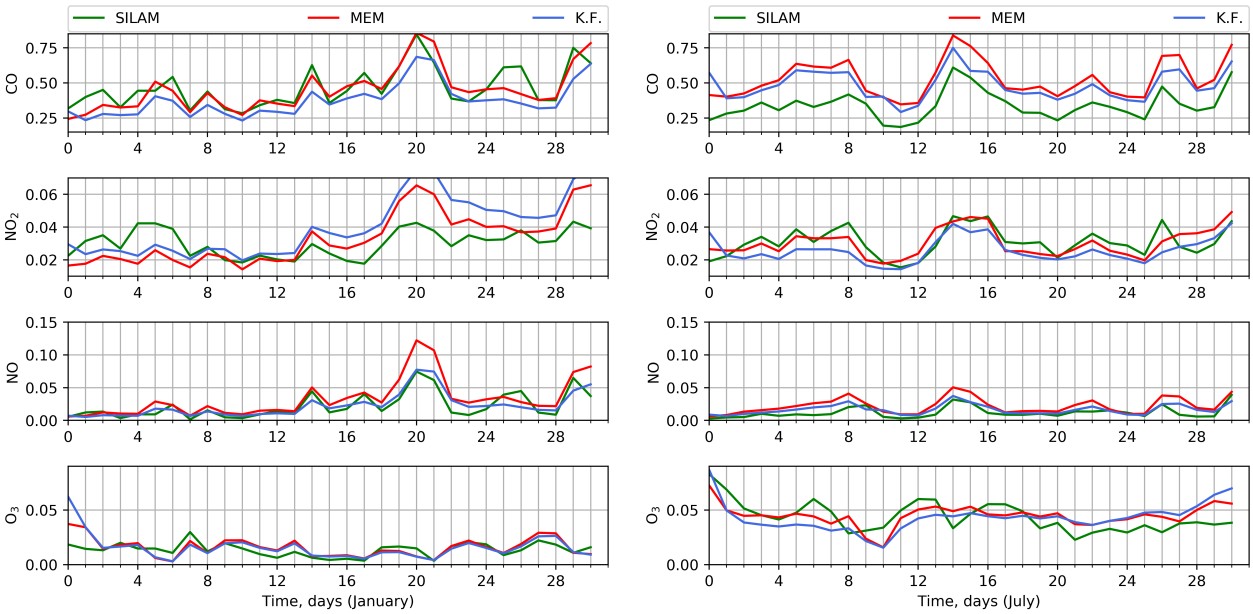

**Figure 7.** The average (over Moscow) diurnal pollutant concentrations according to measurements at the MEM stations (red line) and numerical calculations with the SILAM model (green line) and corrections of these calculations using the Kalman filter (blue line) for the *CO*, *NO*, *NO₂*, and *O₃* concentrations in January and July 2014.

### 4.7. Post-Processing of the Modeled Concentrations

The value of the average intradiurnal correction (Figure 5) and the slowly varying coefficients of the interdiurnal correction (Figure 7) determine the correcting coefficients $k_1$ and $k_2$ for model forecasts with the SILAM model. In order to obtain the corrected SILAM series values $C_{72 \cdot i+j}$ by applying the coefficients $k_1$ and $k_2$, we used the following expressions:

$$C_{72 \cdot i+j} = \frac{\langle q_S \rangle_{72 \cdot i+j} \cdot k_2[i]}{k_1[j]} \quad , \text{ where } \quad k_1[j] = \overline{\left( \frac{\langle q_S \rangle}{\langle q_M \rangle} \right)_j} \text{ and } k_2[i] = \overline{\frac{\langle q_{KF} \rangle}{\langle q_S \rangle}}_i \tag{5}$$

Here, $i = \overline{0, 30}$ is the number corresponding to the day of the month (respective coefficients were calculated for every month), $j = \overline{1, 72}$ is the number of 20 min intervals per day. The $\langle \rangle$ operator means that the values were averaged over all stations and the overline denotes average values for the entire day. The SILAM data series is $q_S$, and $q_{KF}$ represents the values obtained after the Kalman filtering method was applied to the daily mean concentrations.

Figure 8 displays the initial series of diurnal forecasts according to the SILAM model, the corrected (multiplied by the corresponding coefficients $k_1$ and $k_2$ of intradiurnal and interdiurnal corrections) SILAM series, and the observational (MEM) data. As seen from Figure 8 and Table 4, taking into account the temporal relations of model errors allows one to significantly decrease the systematic error. A large portion of the maximum and minimum corrected SILAM series corresponds to observational results for all pollutants in both the summer and winter periods. Moreover, for the summer period, the dynamic statistical error correction model allowed us to correct the amplitude and the phase differences between observations and calculations of the $CO$, $NO$, $NO_2$, and $O_3$ concentrations.

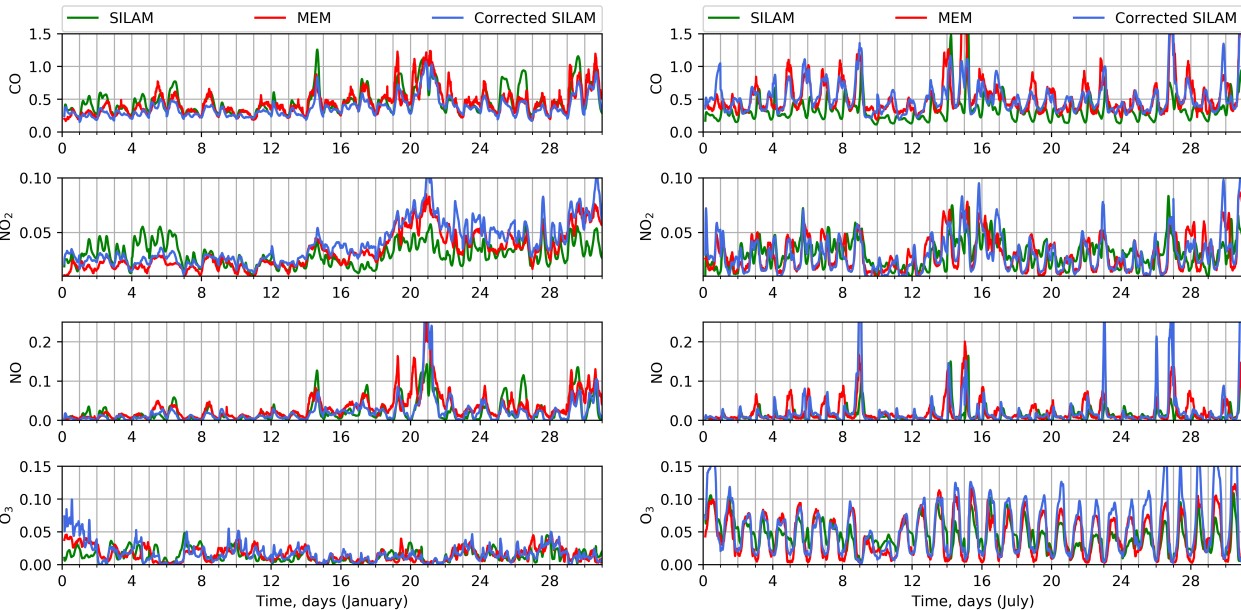

**Figure 8.** Results obtained from Kalman filtering and the intradiurnal correction for the next-day forecast of the average (over Moscow) $CO$, $NO$, $NO_2$, and $O_3$ concentrations (blue curve) according to SILAM data (green curve) and measurement data obtained at the MEM stations (red curve) for January (top) and July (bottom) 2014.

Of course, the purpose of our work is not just a simple correction of the model data by two factors: a constant factor $k_1$ and the dynamic factor $k_2$. Our intention is to conduct a detailed analysis of uncertainties contained in the numerical transport chemical model. A demonstration of the properties of the errors shows that the elementary assumption about their randomness (independence) and the constancy of statistical characteristics is

not correct. Taking into account the spectral properties of errors, according to the results of comparison with measurement data, allows us to conduct post-processing analysis of the results obtained with the working model at the stage of its practical use by "non-specialists" in numerical modeling. The process of model improvement may last for years so that the shortcomings of the "current" version should not be an obstacle to its use "here and now". The proposed approach can find wide application in the analysis of errors of similar models in their practical use.

**Table 4.** Standard statistics values: MB—mean bias, NMB—normalized mean bias, ME—mean error, NME—normalized mean error, Corr—correlation coefficient, for results obtained from Kalman filtering and the intradiurnal correction for the next-day forecast of the average (over Moscow) $CO$, $NO$, $NO_2$, and $O_3$ modeled (SILAM) and corrected concentrations.

| Pollutant | January, 2014 | | | | | July, 2014 | | | | |
|---|---|---|---|---|---|---|---|---|---|---|
| | **MB** | **NMB** | **ME** | **NME** | **Corr** | **MB** | **NMB** | **ME** | **NME** | **Corr** |
| $CO$, SILAM | 0.014 | 0.03 | 0.010 | 0.22 | 0.75 | −0.19 | −0.35 | 0.22 | 0.42 | 0.33 |
| $CO$, Corrected | −0.009 | −0.02 | 0.010 | 0.20 | 0.86 | 0.007 | 0.012 | 0.12 | 0.22 | 0.79 |
| $NO_2$, SILAM | −0.003 | −0.08 | 0.011 | 0.31 | 0.56 | 0.0007 | 0.022 | 0.013 | 0.45 | 0.42 |
| $NO_2$, Corrected | 0.002 | 0.07 | 0.007 | 0.21 | 0.94 | −0.0003 | −0.009 | 0.007 | 0.22 | 0.84 |
| $NO$, SILAM | −0.011 | −0.33 | 0.019 | 0.57 | 0.67 | −0.010 | −0.45 | 0.018 | 0.69 | 0.32 |
| $NO$, Corrected | −0.010 | −0.32 | 0.015 | 0.46 | 0.82 | 0.001 | 0.06 | 0.008 | 0.35 | 0.64 |
| $O_3$, SILAM | −0.002 | −0.12 | 0.008 | 0.51 | 0.42 | −0.001 | −0.03 | 0.02 | 0.39 | 0.71 |
| $O_3$, Corrected | 0.001 | 0.07 | 0.007 | 0.44 | 0.68 | 0.003 | 0.08 | 0.02 | 0.34 | 0.89 |

## 5. Conclusions

Data on pollutant concentrations regularly measured at the MEM network and calculated using the SILAM chemical transport model were compared in detail. It is shown that the spatial correlation scale of measured concentrations is considerably smaller than that of the SILAM spatial grid. The model concentration fields are smooth in time and space, which is explained by low resolution of the model at the street-level description, the lack of detailed data, and the random nature of sources and sinks of the pollutants. For the territory of Moscow, it was rather difficult to single out a general spatial pattern of variations in calculation and measurement errors. Stations located sufficiently close to one another inside the city may yield significantly different results, while model values produce similar results. For the stations located outside the city (Zvenigorod, Zelenograd, and Troitsk), the model often underestimates pollutant concentrations.

The concentrations of the long-lived $CO$ and $NO_2$ pollutants show the best agreement for both diurnal and day-to-day variations. For these pollutants, the results of numerical simulation and the previously estimated urban emissions [10] may be regarded as satisfactory. The diurnal harmonics of the CO and $NO_2$ concentrations are reproduced by the model with an accuracy of about 20% for January 2014 and 40% for July 2014. The diurnal dynamics of the average (over the city) $NO$ and $O_3$ concentrations is accurately described for the daytime and during rush hours. The errors noticeably increase in the night-time, when the model significantly underestimates the $NO$ concentration and overestimates the $O_3$ concentration.

The probability distribution functions constructed for the ratio between the measured and calculated concentrations of each pollutant for January and July 2014 allow one to visualize the frequency of model over- and under-estimation of the average pollutant concentrations. The chemically active $NO$ and $O_3$ pollutants demonstrate higher differences. The distribution tails, i.e., the recurrence of small values of the relative error $\langle q_S \rangle / \langle q_M \rangle$, noticeably increase for these pollutants in July. It can be suggested based on the analysis of

diurnal variations in the ratio between the model and measured concentrations that such a situation is usually observed at night.

The relative errors of the diurnal cycle of ozone and the errors in estimating its average diurnal concentration are significantly larger in summer, which may be associated with the blooming period and an increased emissions of volatile organic compounds of biological origin (Bio-VOCs), whose emission can be specified rather approximately in the current version of the model. Moreover, the uncertainties in describing the diurnal cycle of ozone affect the nitrogen oxides due to the photochemical interaction between these pollutants.

Correcting the lower frequency interdiurnal component of uncertainties may be regarded as Kalman filtering that uses the knowledge of correlation properties of model errors for correcting the numerical forecasts. The forecast correction according to the current (20 min) measurement data makes it possible to significantly decrease a systematic error of the model, which is associated with uncertainties in sources and turbulent transport characteristics and based on comparisons between numerical calculations and measurement data for the previous day, to obtain information sufficient for the correction of diurnal forecasts of average (over the city) concentrations of the key pollutants. Of course, this error may also be reduced by improving the model.

In conclusion, we would like to draw the reader's attention to the fact that we have demonstrated only the first steps towards the analysis and correction of model errors. It is important to mention that the focus of the presented work is on diagnostics, not on forecasting errors. Some further statistical analysis of errors in numerical simulations and in classification of the observation stations by data quality, time periods of their stable operation, the environment type and respective micrometeorological features will allow one to develop and improve the obtained *error model* for numerical forecasts of atmospheric pollution over the Moscow megacity. Analyzing the statistical properties of errors in such numerical forecasts and their dependence on the location of measurement sites (highways, industrial, residential, and recreation zones), one will be able to improve such models and the procedure of chemical transport model and monitoring data fusion for correcting numerical calculations.

**Author Contributions:** Conceptualization, N.P. and V.Y.; methodology, N.P. and V.Y.; software, N.P.; validation, N.P.; formal analysis, N.P., V.Y. and N.E.; investigation, N.P. and V.Y.; resources, N.E.; data curation, N.P.; writing—original draft preparation, N.P., V.Y. and N.E.; writing—review and editing, N.P., V.Y. and N.E.; visualization, N.P.; supervision, V.Y.; project administration, N.E.; funding acquisition, V.Y. and N.E. All authors have read and agreed to the published version of the manuscript.

**Funding:** The collection and analysis of observation data were carried out with the support of the Russian Science Foundation (RSF) (project 16-17-10275) and the Russian Foundation for Basic Research (RFBR) (project 17-29-05102). The methodology for calculating pollutants' temporal variations and emissions was developed in the framework of the RFBR (project 19-05-50088 and 18-29-10080). The development of the statistical error model for urban pollution in Moscow was made in the framework of RFBR (projects 19-35-90073 and 19-05-00028).

**Institutional Review Board Statement:** Not applicable.

**Informed Consent Statement:** Not applicable.

**Data Availability Statement:** Restrictions apply to the availability of these data. Data was obtained from The State Environmental Protection Institution "MosEcoMonitoring" and are available online at https://www.mosecom.ru (accessed on 12 March 2021) with the permission of "MosEcoMonitoring".

**Acknowledgments:** The authors greatly appreciate the constructive comments and helpful discussions with M. A. Sofiev and R. D. Kouznetsov, and they wish to express their gratitude for the provision of original codes of the SILAM model and data on the boundary and initial conditions. We also thank the management and the staff of the Moscow Ecological Monitoring network for providing observational data.

**Conflicts of Interest:** The authors declare no conflict of interest.

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
