# Peer review of "Air Pollution in Moscow Megacity: Data Fusion of the Chemical Transport Model and Observational Network"

_atmosphere, doi:10.3390/atmos12030374_

Round 1

Reviewer 1 Report

A review on the manuscript in Journal Atmosphere entitled „Air Pollution in Moscow Megacity: Data Fusion of the Chemical-Transport Model and Observational Network“.

The article compares air pollution measurement data obtained at the Moscow Ecological Monitoring network (MEM) with numerical simulations using a chemical transformation and transport model (SILAM) in the Moscow Megacity. The analysis showed that the simulation errors are significant. The difference between the calculated and measured concentrations can be up to 100%.

Broad comments

The research methods and results are adequately described. The conclusions stem from the analysis.

Unfortunately, the article does not thoroughly analyze the reasons for the difference in results, either from the inadequate treatment of meteorological parameters in the dispersion model or from the uncertainty of the pollutant emission data and the spatial-temporal dynamics of pollutant emissions.

Specific comments

The article needs minor technical revision.

The links in the article to the various Internet sources (lines 118, 123, 128, 130, 142) should be provided in the Reference List, together with the date of use.

The article cites Figure 3 first (line 125), then Figure 1 (line 213), then Figure 4 (line 223), then Figure 5 (line 227), followed by Figure 2 (line 238). All Figures and Tables should be inserted into the main text close to their first citation and must be numbered following their number of appearance (Figure 1, Figure 2, Table 1, etc.).

Referring to Figures should be in the same style throughout the article - but line 125 uses "Fig.3", line 172 uses "Fig. 3", and line 357 uses the third variant "Figure 5", etc.

There could be a dot after the table number (lines 142+, 253+, 314+) not a colon.

Author Response

Response to the Reviewer.

We would like to thank the reviewer for the positive and constructive feedback, which helped to improve the quality of the article. The Reviewer pointed out to the important question about the cause of the discrepancy between modelled and measured concentrations. In the introduction of our article we mention some of the main causes (the urban emissions, improper model parameterizations, and the measurement errors), which lead to the difference in results. Later in the paper we describe our proposed method, which allows one to separate long-term quasi-systematic errors from the short-term ones. The problem is that different types of error sources can simultaneously cause long-term errors. A simple comparison of model errors with meteorological parameters (wind speed, boundary layer height, temperature) allows us to relate some peaks in errors for a long-lived CO to the low values of wind speed and boundary layer height. So, one of the main reasons for such errors in modelled carbon monoxide concentrations may be the model parametrization of subgrid mixing and mesoscale wind speed fluctuations. However, these parameters do not explain all the differences and in the case of more chemically reactive nitrogen oxides and ozone there is no longer strong correlation between them and model errors. We think that variations in model errors are definitely connected to the unknown changes and spatial heterogeneity of anthropogenic pollutant sources, as well as parameterizations used in the chemical block and approximate values of turbulent diffusion coefficients. The solution to all these potential problems will require a separate study.

Response to Specific comments.

The article needs minor technical revision.

The links in the article to the various Internet sources (lines 118, 123, 128, 130, 142) should be provided in the Reference List, together with the date of use.

The following links have been re-checked, corrected and added to the Reference list:

24. The High Resolution Limited Area Model. Available online: http://hirlam.org (accessed on 01.03.2021).

25. System for Integrated modeLling of Atmospheric coMposition (SILAM). Available online: https://silam.fmi.fi/thredds/catalog (accessed on 01.03.2021).

26. The TNO-2011 Inventory data. Available online: http://www.tno.nl/emissions (accessed on 01.03.2021).

29. Emissions of atmospheric Compounds and Compilation of Ancillary Data (ECCAD). Available online: https://eccad3.sedoo.fr (accessed on 01.03.2021).

30. The State Environmental Protection Institution “MosEcoMonitoring”. Available online: https://www.mosecom.ru (accessed on 01.03.2021).

The article cites Figure 3 first (line 125), then Figure 1 (line 213), then Figure 4 (line 223), then Figure 5 (line 227), followed by Figure 2 (line 238). All Figures and Tables should be inserted into the main text close to their first citation and must be numbered following their number of appearance (Figure 1, Figure 2, Table 1, etc.).

All Figures and Tables are now placed near their first citation and numbered following their appearance.

Referring to Figures should be in the same style throughout the article - but line 125 uses "Fig.3", line 172 uses "Fig. 3", and line 357 uses the third variant "Figure 5", etc.

Changes were applied according to the Reviewer’s suggestion to keep all references to Figures in the same style:

 Reference to the Fig. 3 on the line 125 has been removed.

“Figure 5” on the line 357 is replaced by “Fig. 5”

There could be a dot after the table number (lines 142+, 253+, 314+) not a colon.

We decided to keep colons after table numbers (lines 142+, 253+, 314+)

Reviewer 2 Report

The manuscript compares chemical transport model predictions with measurements of several pollutants in the Moscow region over January and July. The authors evaluate the model/measurement agreement using several statistical measures and then use statistical techniques to improve the model predictions to produce a more accurate forecast. 

As they are written, the conclusions are not particularly useful for a general audience. I seems that this is more of a methods paper as the model evaluation only uses two months of data, but it is unclear how the reader might apply these methods to identify the causes of poor model performance and ultimately improve the model. In my opinion, these flaws are not “fatal”, and some needed additional analysis and commentary can dramatically increase the usefulness of the manuscript.

Major comments

Bias correction of model output is a common way of significantly improving model performance. Typically, for a prediction day, the model searches for days with similar meteorology or emissions in the past and uses the model/measurement biases on those days to correct the prediction. Since this is such a commonly used and successful technique, there should be a discussion of this in the introduction and the authors should compare the pros and cons of this technique relative to what they propose in the manuscript.

It appears that the Kalman extrapolation and correction procedures do improve model performance based on Figures 7 and 8. However, the authors should calculate some model performance statistics such as root mean squared error or average bias with the uncorrected and corrected model data to quantify the improvements in accuracy. An evaluation of the statistical significance of the improvements would also be useful. In addition, it looks as if the Kalman filter was fit with the same data that was ultimately corrected. It may be difficult with only two months of data, but it would be best to fit the filter parameters with a hold-out data set to ensure that the correction is not over-fit.  

There is a large disconnect for how a reader might use these techniques outlined in this paper to help improve a chemical transport model. I surmise that this is the key reason why someone would want to read this paper. Perhaps the authors should draw some preliminary conclusions on what areas of the model should be improved based on the statistical approach outlined in the paper. The authors should also provide a framework for how to interpret the spectral properties of the errors more generally so others may use these techniques for other models.

Minor comments

Line 110: the phrase “large number of chemical bonds” is not really clear. Does this mean that the gas-phase species are generally large compounds and are not well-parameterized in the model?

Figure 3: it is not clear what the rectangle means in each of these figures

Line 364: “to” is missing between “difficult” and “estimate”

Figure 7: this figure would be very hard to read for people that are colorblind. I recommend changing the color scheme so that it is more colorblind friendly and/or changing the line styles so they can be distinguishable without color.

Author Response

Response to the Reviewer.

We are grateful to the Reviewer for their insightful comments on our paper. Indeed, we demonstrated only the first steps in the direction of analysis and correction of model errors. We will continue working on model error analysis. Our next work probably will be related to the impact of the 2020 lockdown on atmospheric emissions where will use a bigger sample size than just two months.

 Although the Reviewer rightly notes that the bias correction method is a standard procedure, our work focuses more on the fact that the model deviations do not remain constant, but have a wide range of spatial and temporal scales. The study of these spatial and temporal properties of errors is the aim of the work. Thus, the main focus of the presented paper is on diagnostics, not on forecasting errors. In the conclusions section, a comment has been added: “In conclusion, we would like to draw the reader’s attention to the fact that we have demonstrated only the first steps towards the analysis and correction of model errors. It is important to mention that the focus of the presented work is on diagnostics, not on forecasting errors.” (Lines 514 -517). One of the possible problems with the approach of searching days with similar meteorology is that data from previous years are usually stored with a lower temporal resolution, which can lead to additional errors. We absolutely agree with the Reviewer that it would be interesting to compare the results of our method with the most widespread ones, including bias correction.

The comments of the Reviewer about the need to calculate error statistics after correction are, of course, true. The authors acknowledge the imperfection of the presented work, and due to lack of time, hope that these flaws are not “fatal”. The authors will certainly take this note into account in future work. We have added in Section 4.6 an additional Table 4 (Line 455+) containing standard statistics parameters for the comparisons shown in the Figure 8.

The question of improving a specific chemical-transport model is not at all simple. This process is still ongoing in close cooperation with the developers of the SILAM model. At the same time, the assessment of the spectral properties of errors, as stated in the work, seems to be a separate line of research. But of course, as the reviewer rightly points out, some framework is needed to interact with fashion designers, like perhaps the CMIP-6 project. The Reviewer's suggestions are in line with the further work of the authors.

Response to the Minor comments:

Line 110: the phrase “large number of chemical bonds” is not really clear. Does this mean that the gas-phase species are generally large compounds and are not well-parameterized in the model?

We meant that a large number of chemical bonds between the modeled components leads to a large number of chemical interactions which are only approximately parameterized.

Figure 3: it is not clear what the rectangle means in each of these figures

The caption to Figure 3 has been changed to clarify the meaning of the rectangle.

Line 364: “to” is missing between “difficult” and “estimate”

“to” has been added on the line 364.

Figure 7: this figure would be very hard to read for people that are colorblind. I recommend changing the color scheme so that it is more colorblind friendly and/or changing the line styles so they can be distinguishable without color.

The results of Figure 7 are used to obtain curves shown in the Figure 8. We have added Table 4 to quantify the correction results.

Reviewer 3 Report

This study describes the application of the SILAM chemical transport model over Moscow.  The authors thoroughly evaluate the model against available observations and provide insights on model capabilities by decomposing and analyzing the errors.  The authors then describe methods for correcting errors for forecasting applications.  Overall this is a worthwhile study and provides insights on the limitations of the state of modeling for the Moscow Megacity.

Minor Comments:

Line 98: the CB-4 chemical mechanism is quite old.  Carbon Bond is up to at least revision 3 of CB6.  Is this the most recent mechanism in SILAM?

Line 114-115: I recommend putting the approximate resolution in km in parenthesis after the resolution degrees is specified

Line 124-127: I recommend providing slightly more detail on the model emissions.  Are emissions developed for individual sectors with day-specific inputs?

Line 192: Should “sinc” be “sine”?

Figure 1: Can standard statistics for normalized and absolute bias and error and correlation be provided on the figure?

Figure 1: Why was the Marino station selected for the figure? 

Figure 1 caption: The caption indicates that errors are shown, but strictly speaking concentrations for the model and measurements are being compared.

Figure 2: Why does the model have such a steep S-curve compared with the observations?

Figure 3: The authors could consider making this plot with a base map for the background that shows relevant features such as roadways, etc.  Could also consider zooming in on the map to see the urban features, perhaps at the expense of dropping a few remote sites.  Please describe the meaning of the blue box in the caption.

Figure 5: what drives the bimodal pattern for many species?  Is this associated with traffic patterns or perhaps the diurnal evolution of the boundary layer?

Author Response

Response to the Reviewer.

The authors are grateful to the Reviewer for the detailed analysis of our work and his kind comments. Below we address each specific issue and the manuscript has been updated accordingly.

Response to Minor Comments:

Line 98: the CB-4 chemical mechanism is quite old.  Carbon Bond is up to at least revision 3 of CB6.  Is this the most recent mechanism in SILAM?

We consulted with the model developers about this issue and according to them, the chemical block of the model was originally created on the basis of CB 4, but later in subsequent years it was heavily modified and today the model uses 8 chemico-physical transformation modules (basic acid chemistry and secondary aerosol formation, ozone formation in the troposphere and the stratosphere, radioactive decay, aerosol dynamics in the air, pollen transformations). The description of the model in the text has been corrected accordingly.

Lines 98-99: “… the DMAT and CB-4 (carbon-bond version 4) blocks [20]. The DMAT block allows one to define both gas-phase and heterogeneous interactions in the atmosphere.”

were replaced by “8 chemico-physical transformation modules, which describe basic acid chemistry and secondary aerosol formation, ozone formation in the troposphere and the stratosphere, radioactive decay, aerosol dynamics in the air, pollen transformations [20].”

Line 114-115: I recommend putting the approximate resolution in km in parenthesis after the resolution degrees is specified

The approximate model grid resolution in km has been added on the line 115.

Line 124-127: I recommend providing slightly more detail on the model emissions.  Are emissions developed for individual sectors with day-specific inputs?

We used TNO-2011 inventory data on emissions for most pollutants with exception for CO and NOx. To set up CO and NOx emissions, we used the estimates for annual emissions obtained in one of our previous works (https://doi.org/10.1016/j.atmosenv.2017.11.057), which were then distributed over time with 1h time step using TNO—2011 temporal emission profiles. To distribute the emissions across the Moscow territory first we set the share of total megacity emission to each model grid using several parameters derived from air pollutant concentrations distribution and locations of urban sources. Then varying spatial distribution parameters, we minimized the model errors. The emissions thus obtained were then verified in numerical experiments with two chemical-transport models in work (DOI: 10.1134/S1024856020040090).

Lines 124-128: “Both the CO and NOx emissions within the territory of the Moscow megacity, whose area is marked by blue rectangle in Fig.3, were specified based upon analysis of data obtained at the MEM network and from numerical experiments on the optimization of urban air-pollution sources (see [10,24,25]). The CO and NOx emissions outside Moscow were taken from the TNO-2011 Inventory data (http://www.tno.nl/emissions).”

were replaced by: “Most air pollutant emissions were taken from TNO-2011 Inventory data (http://www.tno.nl/emissions). The CO and NOx emissions within the territory of the Moscow megacity were specified based upon annual emission values provided in [10]. Analysis of data obtained at the MEM network and numerical experiments on the optimization of urban air-pollution sources described in [24, 25] allowed us to distribute them over time with 1-hour time step and across Moscow territory. The CO and NOx emissions outside Moscow were taken from the TNO-2011 Inventory data.”

Line 192: Should “sinc” be “sine”?

The use of “sinc” function on the line 192 is correct because spectral transmission-window for simple moving-average is also “sinc”.

Figure 1: Can standard statistics for normalized and absolute bias and error and correlation be provided on the figure?

Figure 1 was changed accordingly. Now it shows standard statistics for the displayed time series.

Figure 1: Why was the Marino station selected for the figure? 

We chose this station as a typical example of a measured time series, which allows us to demonstrate that measured concentrations averaged over megacity show a significantly better agreement with model than those before averaging.

Figure 1 caption: The caption indicates that errors are shown, but strictly speaking concentrations for the model and measurements are being compared.

Figure 1 caption has been changed to:

“Local errors and the errors of average concentrations (MB – Mean Bias, NMB – Normalised Mean Bias, ME – Mean Error, NME – Normalised Mean Error, corr- Correlation coefficient). Top: the NO2 concentrations measured on July 1-31, 2014, and calculated using the SILAM model for the same period for the Mar’ino station. Down: the NO2 concentrations averaged over the selected MEM stations for a period of July 1-31, 2014, according to measurement data and numerical calculations.”

Figure 2: Why does the model have such a steep S-curve compared with the observations?

It is one of our main conclusions that the spatial correlation scale of measured concentrations is really small as compared to the SILAM spatial grid. The model concentration fields are smooth in time and space meaning that their deviations from the average (over Moscow) concentration values <q> are significantly smaller than those for measurements and the cumulative distribution S-curve for normalised concentration values q/<q> is steeper. This fact can be explained by low resolution of the model at the street-level description, the lack of detailed data, and the random nature of sources and sinks of the pollutants.

Figure 3: The authors could consider making this plot with a base map for the background that shows relevant features such as roadways, etc.  Could also consider zooming in on the map to see the urban features, perhaps at the expense of dropping a few remote sites.  Please describe the meaning of the blue box in the caption.

The Figure 3 has been zoomed in and the meaning of the blue box has been added to the caption:

Line 290+: “Figure 3. Spatial distribution of statistics of calculated and measured (at the MEM network) concentrations of CO, NO, NO2, and O3 for January 1-31 (top row) and July 1-31, 2014 (bottom row). The characteristic value of the relative interquartile range Qr at each station is denoted by marker types (triangle, circle, and square). The area for which the CO and NOx emissions were specified based upon analysis of data obtained at the MEM network and from numerical experiments on the optimization of urban air-pollution sources [10,24,25] is marked by blue rectangle.”

Figure 5: what drives the bimodal pattern for many species?  Is this associated with traffic patterns or perhaps the diurnal evolution of the boundary layer?

We think that both factors are likely to affect the results. Error maxima coincide with the morning and evening peaks of traffic activity. Also, comparisons of the model errors time series with the data of meteorological parameters (Boundary Layer Height (BLH), wind speed) show that model errors increase significantly in the presence of early morning temperature inversions and atmospheric blockings when BLH and wind speed values are low.

Reviewer 4 Report

Dear authors

you have really done a huge work in your comparisons, and do a real first step in the good direction when you use quotients to compare results. Also the use of Kalman filtering, that is geostatistical models, to implement spatial variability is a good issue.

Nevertheless, I think you do a great error: you calculate mean values using arithmetic averages. In fact, concentrations are NOT real values, they are part of a total, percentages or ppm... The central mean for this kind of data is the geometric mean.

Please, read this paper: "2009 Peter Filzmoser, Karel Hron, Clemens Reimann Univariate statistical analysis of environmental (compositional) data: Problems and possibilities Science of the Total Environment 407  6100–6108 doi:10.1016/j.scitotenv.2009.08.008"

I think that you should improve your work improving the mathematical issue. Then, your work will be really sound.

Author Response

Response to the Reviewer.

We want to especially thank the Reviewer. His comments on the significance of the submitted work are greatly appreciated. The reviewer absolutely rightly points out that the geometric mean of concentrations is of great importance when comparing the results of chemical-transport modeling and data from local observations. This approach was carefully analyzed and actively used by the authors during the writing of the manuscript. Figures 2,4,6, as can be seen, are presented on a logarithmic scale, Tables 2 and 3 show values in relative terms. The mean is also used to normalize logarithms. Recognizing this, the authors chose the way of presentation more understandable for the reader and "hidden" part of the mathematical reasoning. We gratefully accept a link to an interesting paper ("2009 Peter Filzmoser, Karel Hron, Clemens Reimann Univariate statistical analysis of environmental (compositional) data: Problems and possibilities Science of the Total Environment 407 6100–6108 doi: 10.1016 / j.scitotenv.2009.08 .008 ").

Round 2

Reviewer 2 Report

The authors addressed my major concerns in the revised manuscript.